# Nurses’ Perspectives on Smoking Policies, Safety and Cessation Support in Psychiatric Wards: A Cross-Sectional Survey

**DOI:** 10.3390/healthcare10091735

**Published:** 2022-09-09

**Authors:** Ewelina Chawłowska, Monika Karasiewicz, Katarzyna Marcinkowska, Bogusz Giernaś, Paulina Jóźwiak, Agnieszka Lipiak

**Affiliations:** Department of Preventive Medicine, Poznan University of Medical Sciences, 60-781 Poznan, Poland

**Keywords:** smoking, safety, smoking ban, smoke-free policies, smoking-cessation intervention, tobacco treatment, mental health nurses, psychiatric wards, unhealthy behaviours

## Abstract

A high prevalence of smoking and low rates of smoking cessation interventions can be observed in psychiatric wards. A questionnaire-based, cross-sectional study was performed in five hospitals among 107 psychiatric ward nurses. The aim was to investigate nurses’ views on patients’ smoking practices and their influence on the safety of both the patients and medical personnel. In addition, we asked about the availability of smoking cessation support. Most of the respondents noticed the negative impacts of smoking on patients and medical personnel. Nearly a third of our respondents (29.0%) recalled smoking-related accidents in their facilities. In 45.2% of these accidents, a patient set someone else on fire. Around one fifth of nurses had rather permissive attitudes towards tobacco use in hospital wards. Significant associations were identified between respondents’ smoking status and their opinions on amending smoking policies and on unsupervised smoking. Regarding professional help available to smoking patients, 88.8% of participants reported that interventions to address smoking were available in their wards. Psychiatric hospitalisation can be an opportunity to offer tobacco treatment to patients with mental health conditions. To make use of this opportunity, smoke-free policies need to be put in place and hospital personnel, particularly nurses, should be trained and equipped with the knowledge and skills needed to assist in the smoking care of psychiatric ward patients.

## 1. Introduction

Smoking remains a major preventable health risk factor globally [1], imposing a disproportionate burden on persons with mental health conditions [2]. Their use of tobacco products is estimated to be two to three times more prevalent in comparison to that of the general population [3,4]. These individuals are also more likely to suffer from smoking-related conditions such as chronic obstructive pulmonary disease (COPD) [5], cardiovascular disease [6,7] and cancer [7]. Hence, smoking cessation should be particularly promoted in such populations.

It could be expected that a good opportunity for mental health patients to quit smoking in relatively safe conditions arises when they are hospitalised: first, because healthcare facilities are generally required to implement smoke-free policies; second, because healthcare personnel is present to provide patients with professional smoking cessation interventions [2,8,9,10,11,12,13,14,15].

As for the first solution, the adoption of the WHO Framework Convention on Tobacco Control (FCTC) in 2005 [8] and the European Union (EU) Council Recommendation on smoke-free environments in 2009 [9] motivated a number of EU countries to implement legislation aiming to create smoke-free health settings. This included psychiatric hospitals and wards, where smoking bans were recommended to protect smoking patients from the direct effects of tobacco use [10], and also non-smoking patients and health workers from the detrimental effects of second-hand smoke [10,11,12]. Unfortunately, while most healthcare settings are now completely smoke-free in EU Member States, the implementation of similar policies in psychiatric units has turned out to be difficult [16]. One of the barriers might be a long-standing smoking culture pervading these settings, as reflected in the staff’s belief that a smoking ban would result in increased patient agitation [17,18]. As a result, many psychiatric hospitals and wards in the EU are still exempt from smoking bans or employ partial bans effective only indoors [16,17,19].

In Poland, a strict national ban on smoking in public spaces, including healthcare facilities, was introduced in 2010 [13]. Despite the ban, the actual tobacco policies in psychiatric hospitals and wards differs from place to place [20]. The smoking ban was perceived by most of their staff as impractical, if not completely impossible; among other things, it was feared that tobacco-addicted patients would smoke anyway, thus posing a safety risk [20]. Such observations led to the liberalisation of the law and since 2016 it has been possible—but not compulsory—for psychiatric units to set up enclosed smoking rooms, except in units of increased or maximum security, which remain totally smoke-free [14].

Apart from smoke-free policies, psychiatric facilities are also expected to implement effective smoking cessation interventions. Indeed, interventions which can significantly improve physical and mental functioning of psychiatric patients are available [2,21,22], but, unfortunately, they are not offered often enough [23,24]. One reason may be that some psychiatric patients are less likely to express interest in quitting smoking [25]. Another difficulty is that patients with mental health conditions may require tailored interventions in order to improve treatment outcomes [25,26]. Other barriers mentioned by healthcare professionals include perceptions that mental health patients have more immediate health issues [27], that pharmacological smoking cessation would be too costly to implement [20] and that professionals would not be able to make time to treat smoking in their working routines [28].

It seems, then, that the third factor crucial in the effective implementation of smoke-free policies and smoking cessation interventions is the active involvement of mental health personnel [2,29,30,31]. To secure such involvement, professionals working in psychiatric units need to be equipped with up-to-date knowledge regarding policies and interventions, and need to be motivated to use and convincingly promote them. This is why we decided to investigate the knowledge and attitudes of nursing personnel at psychiatric wards regarding patient smoking practices followed in their workplaces. Extensive research on this subject is available [19,28,29,30,31,32,33,34,35,36,37,38,39,40], but few studies were conducted in Poland [20,41] and, to the best of our knowledge, no standardised measures have been developed or adapted and validated for use among Polish psychiatric inpatients. Thus, the aim of our study was to explore the views of nurses working in psychiatric wards on patients’ smoking practices and their influence on the safety and comfort of both the patients and staff. We wanted to examine various aspects of the use of smoking rooms, the care of smoking patients including smoking cessation support, the policies and rules related to patients’ tobacco use, as well as possible safety risks caused by smoking patients. This way, we attempted to find out how psychiatric nurses view the issue of patients’ tobacco use after the 2016 liberalisation of relevant legislation.

## 2. Materials and Methods

We used a self-developed anonymous questionnaire prepared at the Poznan University of Medical Sciences in Poland. We chose to design a scale exploring the perspectives of nurses only, because they are the staff members who are responsible for enforcing smoking policies at wards and at the same time spend the most time with hospitalised patients, which makes them ideally placed for smoking cessation counselling. The questionnaire was developed on the basis of a literature review as well as comments from nurses interviewed prior to the study. The resulting pool of questions was revised by three experts (two with public health and one with nursing background) each having at least 10 years of experience in public health and health interventions. The number of questions was reduced to 14 in an attempt to make the questionnaire short enough to provide higher response rates. The final version of the questionnaire included demographic questions (Table 1) and addressed nurses’ perceptions on two aspects of smoking practices: tobacco use safety and smoking cessation support (Table 2 and Table 3). The safety aspect included questions about the actual patient tobacco use practices followed in participants’ workplaces and the related safety issues (e.g., the availability of a smoking room, supervision of cigarette supply and use, smoking-related accidents, influence of smoking on others), and also explored nurses’ opinions on smoking policies in general (i.e., on allowing patients to smoke and on the national law on tobacco use in psychiatric facilities). The smoking cessation aspect was reflected in a question about the availability of smoking cessation support in participants’ workplaces.

This survey was conducted from 27 September to 20 November 2019, by means of paper-and-pencil interviewing, among nurses employed at the psychiatric wards of five different hospitals in two neighbouring provinces of Western Poland. They were all multi-specialty hospitals with psychiatric wards and represented either the first or the second level out of three levels of hospital care available in Poland (with higher level hospitals offering more specialty wards and more specialised care). The two first level hospitals were located in big cities, while the three second level hospitals were located in smaller towns. Convenience sampling was used. The only inclusion criteria for the participants were (1) working as a nurse at a psychiatric ward and (2) consenting to take part in the study. The questionnaires were made available to potential participants at nurses’ stations in particular psychiatric wards.

Participation in the survey was fully voluntary and anonymous, which was made clear to participants beforehand. Informed consent was obtained from members of the study group. According to the policy of the Bioethics Committee of the Poznan University of Medical Sciences, our study did not constitute a scientific experiment and, as such, did not need ethical approval in accordance with Polish law and Good Clinical Practice.

Descriptive analysis of the collected data was carried out. A chi-squared test was performed using Statistica 13.3.721.0 (TIBCO Software Inc.) to examine relations between nurses’ perceptions and knowledge on the one hand and personnel- and ward-related variables on the other hand. Results below *p* = 0.05 were considered statistically significant.

## 3. Results

### 3.1. Study Group

The study group comprised of 107 participants, mostly female (81.3%), aged below 50 years (86.9%), and with work experience of more than 10 years (75.7%). In the whole group, 27.1% were smokers (29.9% of female and 15.0% of male participants). Over a half (57.0%) of our participants were employed at first level (i.e., less specialised) hospitals. Participants’ demographic characteristics are presented in Table 1.

### 3.2. Smoking Policies and Practices: Safety Issues and Cessation Interventions

According to most respondents (63.6%), the wards they worked at had smoking rooms, 79.4% of which could be accessed without restrictions. In most cases (55.1%) no one supervised smoking patients. It was always the case at wards without smoking rooms and at 29.4% of those with the rooms.

Cigarettes and lighters were usually kept in patients’ bedside cabinets (79.4% and 62.6%, respectively). While 48.6% of respondents indicated that patients were usually provided with cigarettes by family members or friends, 40.2% of respondents did not know the source of their cigarettes.

According to 57.0% of participants, access to cigarettes was not used in their facilities for controlling patients. However, 23.4% said that cigarettes were used for this purpose. Such opinions were voiced only by those respondents who worked at wards with smoking rooms (yes—35.3%, no—45.6% and I have no opinion—19.1%).

Nearly a third of our respondents (29.0%) recalled smoking-related accidents in their facilities. In 45.2% of these accidents, a patient set someone else on fire.

Regarding professional help available to smoking patients, 88.8% of participants reported that interventions to address smoking were available in their wards in the form of group therapy, nicotine replacement treatment or behavioural therapy.

Other details of the smoking policies and practices as reported by the medical personnel are presented in Table 2.

When it came to the medical personnel’s opinions on allowing patients to smoke, 38.3% said that patients should smoke only under staff supervision, while 19.6% believed they should not be allowed to smoke at all. Over a fifth of our respondents (22.4%) felt that patients should be allowed to use tobacco without any supervision. The remaining 19.6% did not have an opinion. Strikingly, 32.7% of participants did not observe any negative influence of smoking patients on non-smoking ones. However, as many as 51.4% did notice the negative impact, either emotional (aggression, anxiety) or behavioural (encouragement to smoke). While 34.6% of respondents did not notice a negative impact of smoking patients on healthcare personnel, the majority did (65.4%) (posing a risk to personnel, making their work harder, requiring extra attention, encouraging personnel to smoke). Nearly a half of the study group (46.7%) believed that smoking patients posed a safety risk in the ward. One in two respondents (54.2%) believed that the law on smoking in psychiatric units should be amended.

### 3.3. Ward Policies and Medical Personnel’s Opinions vs. Selected Ward and Personnel Variables

In our study we also examined relations between selected policies and the medical personnel’s opinions on the one hand, and some ward variables and personnel’s demographics on the other hand. The results are presented in Table 3.

Most of the significant associations we identified were found in relation to hospital level; this variable turned out to be statistically significant in seven out of nine relations we investigated. For example, at second level hospitals, where more specialty wards and more specialised care could be found, smoking rooms were always available, which was not always the case at first level hospitals (*p* = 0.0001). Hospital level also differentiated medical personnel’s opinions regarding smoking policies: participants employed at first level hospitals were more likely to observe the negative influence of smoking patients on non-smoking ones (*p* = 0.01051) and on safety in the ward (*p* = 0.01209). Additionally, they were more likely to report using control over supply of cigarettes (*p* = 0.00828). At second level hospitals there were relatively more supporters of a complete smoking ban, while at first level hospitals more nurses supported smoking supervision (*p* = 0.00659). According to participants, second level hospitals always offered smoking cessation assistance to nicotine-addicted patients, while at first level facilities it was not always available (*p* = 0.00007).

The next variable which was found to affect responses in a number of ways was the occurrence of smoking-related accidents. At the wards where such accidents occurred, respondents were more likely to state that smoking patients posed a safety risk (*p* = 0.02036), to be in favour of a complete smoking ban (*p* = 0.00008), and to support the amendment of smoking-related laws pertaining to psychiatric facilities (*p* = 0.00003).

Finally, we identified a significant association between respondents’ smoking status and their opinions on two issues. Nurses who smoked were less likely to support law amendments (*p* = 0.00442) and more likely to support unsupervised smoking (*p* = 0.02830) than non-smoking nurses.

## 4. Discussion

The problem of smoking in mental health settings has so far been understudied in Poland. Although our study was limited to just a few hospital wards and to nursing staff only, it allowed us to make a few interesting observations regarding psychiatric settings as environments which could become conducive to smoking cessation and, simultaneously, remain safe to both patients and staff.

The prevalence of smoking among psychiatric nurses in our study was 27.1%, and reached 29.9% among female professionals. While the study sample was small, it was notable that the latter proportion is high compared to that of the general population, with 18.9% of Polish females aged 15 years and older estimated to smoke in 2018 [42]. The rate for both sexes among our respondents was also higher than rates reported in other studies conducted among psychiatric nurses, which ranged from 13.4% in China [43], 16% in Australia [36], 17% in New Zealand, 20% in the United States [44], to 22% in the United Kingdom [45]. The rates estimated for Polish nurses of various specialties were 18% in a 2009 study [46], 40% in a 2012 study [47], and 20% in a recent 2021 study [48]. One of the reasons for a higher prevalence of tobacco use in this professional group might be due to work-related stress, which has been suggested to be elevated among nurses [47] and has reached higher levels among mental healthcare staff than in other healthcare specialties [49]. Thus, smoking may be their stress coping strategy [47,50]. On the other hand, high smoking rates in mental health settings constitute a smoking culture with “a long and entrenched history” [17], which affect healthcare professionals through peer pressure, and inevitably influence psychiatric patients through role modelling. It was pointed out that frequent observations of smoking professionals and other patients challenged patients’ own attempts to quit this unhealthy habit [51].

According to our respondents, smoking treatment was not available in all study sites; 11.2% of participants indicated that there were no such services in their workplaces. Although our observations cannot be generalised to all psychiatric facilities, they may serve as a warning signal that the situations should be monitored as some wards might need more support to be able to help patients quit. This is unfortunate, as there are a number of treatment options which have been found to be effective for patients in mental health facilities. For example, Metse et al. (2018) reported that nicotine replacement therapy was effective according to most psychiatric patients who received it [52]. According to Keizer et al. (2019), motivational enhancement interventions were successful in reducing patients’ tobacco withdrawal symptoms [53]. Unfortunately, Prochaska (2011) reported that smoking cessation interventions are still not offered often enough to patients in mental health facilities [54]. A recent review reported that the lack of awareness among healthcare professionals regarding links between tobacco use and mental illness remains a serious barrier to the provision of smoking cessation support [51]. Prochaska (2010) argued that smoking, contrary to professional belief, does not help relieve negative mental health symptoms, and is actually associated with increased depressive symptoms and suicidal risk behaviours [55]. Moreover, it was pointed out in a meta-analysis of 38 studies with 16,369 mental health professionals that 42% of them perceived barriers to such interventions, and 41% had negative attitudes to smoking cessation, with 51% believing that patients are not interested in quitting and 38% were convinced that quitting smoking is too much for patients to take on [37]. Guo et al. (2015) concluded that a serious barrier to the implementation of smoking cessation interventions might be the fact that healthcare professionals may not feel competent enough to conduct them with their patients [38]. Hence, it would be advisable to provide relevant training and develop their competencies, as Prochaska et al. (2017) recommended [2].

An important element of settings and environments which make them more conducive to smoking cessation is the presence of smoke-free policies. Although a smoking ban in psychiatric units is a controversial issue, research suggests that smoke-free psychiatric hospitalisations may reduce the number of cigarettes smoked by nicotine-addicted patients [39,56,57], thus making them more inclined to quit altogether [58], and making them perceive the quitting process as easier [45]. In contrast, the lack of a smoking ban in psychiatric units was found to increase the number of patients who started smoking, probably owing to the influence of hospitalised smokers [40]. Etter and Etter (2007) found that even a partial smoking ban reduced both smoke exposure and annoyance caused by this behaviour among non-smoking patients and staff [59]. According to McNally et al. (2006), it is often psychiatric staff who are reluctant towards adopting smoking restrictions [32]. Our results suggest that about a fifth of our respondents were rather permissive about tobacco use, expressing the view that patients should be allowed to smoke without any supervision, with two groups tending to be more permissive: smokers and nurses employed at less specialised hospitals. We do not have enough data to provide sound explanations for these tendencies, but we could speculate that the more lenient attitude of the latter group might result from weaker institutional and organisational support at less specialised hospitals, reflected, for example, in a lower availability of smoking rooms; this aspect, however, definitely needs more in-depth research. The former finding about smokers representing more permissive attitudes was corroborated in earlier studies. Staff members who smoked were found to be more permissive about tobacco use among patients despite knowing the health risks involved [33]. According to another study, smokers tended to have more reservations about the importance of smoke-free policies and treatment of nicotine dependence among patients [28]. Yet another article reported that smoking nurses were less likely to ask patients about their tobacco use [44] and to advise them to quit [34] than non-smoking ones. Given the cross-sectional nature of our study, it would be difficult to seek causality in the association we found between nurses’ smoking status and their opinions on smoke-free policies. On the one hand, the association might mean that the smoking nurses were reluctant to modify the health-compromising behaviour in patients when they engaged in it themselves. On the other hand, smoking-related culture prevalent in a given setting might have influenced nurses’ smoking status. In fact, Duaso et al. (2016) suggested that the interaction between smoking status and tobacco attitudes may be mediated by tobacco control efforts and acceptability of smoking in a given facility or country [34]. Research mentions yet another reason for staff’s reluctance toward smoke-free mental health settings: it is a fear that tighter restrictions would lead to increased aggression in patients banned from smoking [19]. While staff members were found to report such episodes [18,60], other studies did not confirm any major longstanding negative effects of smoking bans [61,62,63].

Our study also touched upon the topic of smoking-related accidents. It seems that more rigorous attitudes toward smoking patients among medical personnel were often based on previous experiences (smoking-related accidents, bad influence of smokers on both non-smoking patients and staff). Indeed, the available research seems to suggest that an assumption of a connection between stricter smoking policies and fewer accidents is well-grounded. Some of the main reasons for smoking incidents identified by Sohal et al. (2016) included tobacco use itself as a facilitator of undesirable behaviours, certain smoking-related arrangements (e.g., organisation of smoking breaks), and the resulting strains on staff (e.g., the need to enforce restrictions and supervise patients). Consequently, the authors argued that it would be beneficial to make psychiatric settings completely smoke-free [64]. This suggestion was supported by findings from a review of 26 studies, which concluded that the despite staff’s fears, total bans did not increase patients’ aggression, but actually decreased the number of issues [61]. Lastly, an Australian survey found that staff reported patient care to be less challenging after a completely smoke-free environment was implemented [35]. Again, while the scale of our study does not allow for definite generalisations, our findings seem to suggest that the more specialised a facility is, the closer it comes to implementing smoke-free policies, because more specialised hospitals were more likely to have certain smoking-related arrangements (enclosed smoking rooms, smoking cessation support). In addition, their employees were more willing to support total smoking bans. Perhaps, again, stronger organisational support worked better in these settings.

In view of the aspects mentioned above, nursing staff in psychiatric units is crucial and ideally positioned to introducing the change which was addressed in the APNA position statement, that is, in becoming nicotine addiction treatment leaders and innovators in multiple settings, including clinical practice [65]. However, nurses will be reluctant to engage in the provision of cessation services unless they receive organisational support [66], have sufficient knowledge on tobacco dependence [28], and feel more competent in counselling. Unfortunately, as VanDevanter et al. (2007) claimed in their study, curricula are not effectively preparing nursing professionals in this area [67]. What is more, tobacco use seriously affects not only patients, but also nurses themselves: it additionally increases their already elevated cardiovascular risk connected with working in health care [68]. This is why it is crucial to provide smoking cessation support to smokers among nursing personnel as an inherent part of smoke-free policies [69], which is advocated by professional nursing personnel’s organisations [70,71].

## 5. Conclusions

Given the scarcity of similar studies in Poland, the presented results may lead to a better understanding of how the current smoking practices in psychiatric wards may influence smoking safety and support tobacco cessation. Smoke-free health settings are proven to be not only healthier but also safer to patients and staff alike. Unfortunately, the implementation of certain tobacco policies and treatments in hospital premises often requires making decisions which, although evidence-based, may be unpopular and possibly hindered by high prevalence of smoking addiction among psychiatric ward patients and staff, and by permissive personnel attitudes towards smoking.

Hospitalisation in a psychiatric ward can be an opportunity to provide holistic treatment to a patient. One health problem that could be targeted is smoking, which is especially prevalent among and harmful to mentally ill persons. The group which can be particularly significant in providing smoking care is psychiatric nurses, who are not only comprehensively trained health professionals but are also able to deliver context-specific interventions to their patients in mental health facilities. However, engaging nurses in effective smoking cessation treatment is contingent upon three conditions. Firstly, nurses should be equipped with up-to-date knowledge regarding the influence of smoking on health and on psychiatric patients in particular. Secondly, they should receive continuous training in providing smoking cessation support to be able to apply modern, evidence-based methods and approaches and, if necessary, to tailor individual solutions. Last but not least, they should be encouraged and supported to give up smoking if they suffer from tobacco addiction themselves because health professionals acting as role models can become a powerful inspiration towards positive behaviour change and support in patients.

## 6. Study Limitations

This study has its limitations which may have an impact on its generalisability and interpretation. First, the rather small sample size increases the margin of error. Second, it is a self-reported study based on the views and opinions of one group of respondents. We did not explore patients’ perspectives or investigate the actual smoking policies of hospitals. Therefore, the results may be subject to recall bias or reflect participants’ reluctance to report socially undesirable opinions. More in-depth and larger-scale research is necessary to verify associations discussed in the article.

## Figures and Tables

**Table 1 healthcare-10-01735-t001:** Demographic characteristics of study participants (n = 107).

Sex	Female—87 (81%) Male—20 (19%)
Age	20–30 years—30 (28%) 31–40 years—33 (31%) 41–50 years—30 (28%) >50 years—14 (13%)
Work experience	0–10 years—26 (24%) 11–15 years—28 (26%) 16–20 years—11 (10%) 21–25 years—12 (11%) 26–30 years—13 (12%) >30 years—17 (16%)
Employment	First level hospital—61 (57%) Second level hospital—46 (43%)
Smoking status	Non-smoker—78 (73%) Smoker—29 (27%)

**Table 2 healthcare-10-01735-t002:** Smoking policies and practices in psychiatric wards according to nursing personnel (n = 107).

Is there a smoking room in the facility?	Yes—68 (63.6%) No—39 (36.4%)
Is access to the smoking room restricted?	Not applicable—39 (36.4%) Yes—14 (13.1%) No—54 (50.5%)
Who is responsible for supervising patients who smoke?	No one—59 (55.1%) Nurse—24 (22.4%) Nursing assistant—2 (1.9%) Therapist—22 (20.6%)
Where are patients’ cigarettes kept?	In patients’ bedside cabinets—85 (79.4%) In a cabinet in the nurses’ station—22 (20.6%)
Where are patients’ lighters kept?	In patients’ bedside cabinets—67 (62.6%) In a cabinet in the nurses’ station—40 (37.4%)
Who provides cigarettes to nicotine-addicted patients?	I don’t know—43 (40.2%) Health personnel—12 (11.2%) Family or friends—52 (48.6%)
Are cigarettes used for controlling patients?	Yes—25 (23.4%) No—61 (57.0%) I have no opinion—21 (19.6%)
Do nicotine-addicted patients receive any help in the facility?	Yes—95 (88.8%) No—12 (11.2%)
Have there been any smoking-related accidents in the facility?	Yes—31 (29.0%) No—76 (71.0%)
What kind of accidents?	Not applicable—76 (71.0%) Setting oneself on fire—4 (3.7%) Setting fire to another person—14 (13.1%) Setting fire to a building—2 (1.9%) Setting fire to other property—11 (10.3%)

**Table 3 healthcare-10-01735-t003:** Ward policies and personnel’s opinions vs. selected ward and personnel variables.

	Ward Variables				Personnel Variables	
	Hospital Level	Smoking Room	Smoking Cessation Assistance	Smoking-Related Accidents	Work Experience	Smoking Status
**Reported ward policies and practices**						
Is there a smoking room in the facility?	***p* = 0.00001**	X	*p* = 0.07074	*p* = 0.05696	*p* = 0.34637	*p* = 0.79670
Have there been any smoking-related accidents in the facility?	*p* = 0.11386	*p* = 0.05696	***p* = 0.01833**	X	*p* = 0.07332	*p* = 0.24950
Is access to the smoking room restricted?	***p* = 0.00369**	*p* = 1.0000	*p* = 0.85981	*p* = 0.22313	*p* = 0.63427	***p* = 0.00629**
Do nicotine-addicted patients receive any help in the facility?	***p* = 0.00007**	*p* = 0.07074	X	***p* = 0.01833**	***p* = 0.00354**	*p* = 0.37015
**Personnel’s opinions and perceptions**						
How do smoking patients influence non-smoking patients?	***p* = 0.01051**	***p* = 0.03447**	***p* = 0.00001**	*p* = 0.08884	*p* = 0.10812	*p* = 0.08065
How do smoking patients influence safety at the facility?	***p* = 0.01209**	*p* = 0.05461	***p* = 0.00001**	***p* = 0.02036**	***p* = 0.00795**	*p* = 0.47783
Are cigarettes used for controlling patients?	***p* = 0.00828**	***p* = 0.00007**	*p* = 0.07747	*p* = 0.81928	*p* = 0.16071	*p* = 0.50497
What do you think of allowing patients to smoke?	***p* = 0.00659**	*p* = 0.70868	***p* = 0.00639**	***p* = 0.00008**	*p* = 0.23905	***p* = 0.02830**
Do you think that the law on smoking at psychiatric facilities should be changed?	*p* = 0.71062	*p* = 0.91192	***p* = 0.02456**	***p* = 0.00003**	*p* = 0.69191	***p* = 0.00442**

Note: Numbers in bold indicate statistically significant results.

## Data Availability

The datasets generated and analysed during the current study are available from the corresponding author.

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
