# Peer review of "Nurses’ Perspectives on Smoking Policies, Safety and Cessation Support in Psychiatric Wards: A Cross-Sectional Survey"

_healthcare, 2022, doi:10.3390/healthcare10091735_

Round 1

Reviewer 1 Report

I would like to congratulate the authors as I found the research very interesting. I encourage them to expand the results in the future with the opinions of the patients themselves. 

Reviewer 2 Report

in the introduction it should be emphasized whether or not there are instruments in the scientific literature that measure the same phenomenon of study also in the Polish language

The title of the questionnaire and of the study should be changed, because it emphasizes more on safety in terms of fires or accidents within the psychiatric service, rather than policies on smoking as such.

The instrument measures two very different dimensions, one is safety in the presence of accidents such as fire, and the other dimension is education programs to quit smoking, which are not closely related.

In order to be accepted, you must change the title of the article and explain that two different dimensions are being measured.

Reviewer 3 Report

1. Please review the manuscript to correct minor spelling and grammar mistakes. 

2. Please review the missing references in the introduction; see lines 39-41 and 76-78.

3. Materials and Methods section could be improved by including an explanation of the study design, inclusion and exclusion criteria, and recruitment.  In addition, provide the steps taken in developing the questionnaire (provide item examples), the analysis process, and the explanations for these steps. 

4. Overall, the discussion could be improved by a more critical discussion of the implications of the study’s findings, especially concerning the study's cross-sectional nature and the potential bi-directional associations between respondents smoking status and their opinions on amending policies on unsupervised smoking. 
